# Lutein derived from *Xenostegia tridentata* exhibits anticancer activities against A549 lung cancer cells *via* hyaluronidase inhibition

**Jaruwan Chatwichien**[1,2,3]*, **Natthawat Semakul**[4☯], **Saranphong Yimklan**[4☯], **Nutchapong Suwanwong**[1☯], **Prakansi Naksing**[1], **Somsak Ruchirawat**[1,3,5]

**1** Chulabhorn Graduate Institute, Program in Chemical Sciences, Bangkok, Thailand, **2** Chulabhorn Royal Academy, Bangkok, Thailand, **3** Center of Excellence on Environmental Health and Toxicology (EHT), OPS, MHESI, Bangkok, Thailand, **4** Faculty of Science, Department of Chemistry, Chiang Mai University, Chiang Mai, Thailand, **5** Laboratory of Medicinal Chemistry, Chulabhorn Research Institute, Bangkok, Thailand

☯ These authors contributed equally to this work.
* jaruwanc@cgi.ac.th

**Data Availability Statement:** All relevant data are within the manuscript and its Supporting Information files.

## Abstract

Hyaluronidase has been emerging as a potential target for cancer treatment. Herein, the anticancer effects against A549 NSCLC cells and hyaluronidase inhibitory activity of the ethanol extract of *Xenostegia tridentata* (L.) D.F. Austin & Staples and its subfractions were investigated. In correlation with their hyaluronidase inhibition, the hexane subfraction exhibited the most potent cytotoxicity, and the ethyl acetate subfraction could significantly inhibit the cancer cell migration. The hexane and ethyl acetate fractions were then further isolated to identify the active compounds responsible for the anticancer and hyaluronidase inhibitory activities. Among the 10 isolated compounds, lutein (**5**), a previously reported anti-lung cancer agent, showed the strongest inhibition on hyaluronidase enzyme activity. Its anticancer activities were validated. Notably, in addition to demonstrating the potential of *X. tridentata* extract for NSCLC treatment, this study discloses that hyaluronidase is a potential target for the anticancer activities of lutein. The cellular mechanisms underlying the hyaluronidase inhibitory activity of *X. tridentata* extract need to be further explored to fully understand how this inhibition contributes to its anti-cancer effects.

## 1. Introduction

Lung cancer is one of the most common cancer types with a low 5-year survival rate [1]. Due to the increasing air pollution and infectious respiratory diseases, the global number of new patients has kept increasing significantly every year [2]. Since non-small-cell lung cancer (NSCLC) is the major type (approximately 85%) of lung cancer, searching for effective NSCLC treatment and prevention is urgently needed.

Hyaluronan (HA), an acidic polysaccharide composed of different numbers of repeating disaccharide units of D-glucuronic acid (GlcA) and N-acetyl-D-glucosamine (GlcNAc) linked with glycosidic bonds, is one of the major nonprotein components of the extracellular matrix

**Funding:** JC: Office of the Permanent Secretary, Ministry of Higher Education, Science, Research and Innovation (OPS-MHESI) [grant number RGNS 64-239] and Chulabhorn Graduate Institute JC: Chulabhorn Royal Academy (Fundamental Fund: fiscal year 2024 by National Science Research and Innovation Fund [FRB670024/0240 Project code 198480]) NS: the Postdoctoral Research Fund from Chulabhorn Graduate Institute [grant number CGIP(2022)/ 01].

**Competing interests:** The authors have declared that no competing interests exist.

(ECM) of animals [3]. Different sizes of HA are synthesized by the transmembrane enzymes, HA synthases (HAS1-3), and extruded from the cytoplasm to the ECM. Conversely, the polymer chain of HA can be degraded mainly by two factors which are oxygen free radicals and hyaluronidases. Hyal1-2, PH-20, HYBID (hyaluronan binding protein involved in hyaluronan depolymerization), and TMEM2 (transmembrane protein 2) have been identified as hyaluronidases involved in hyaluronan degradation in human tissues [4]. Depending on the size, HA interacts with its receptors differently, leading to dissimilar biological effects [5,6]. CD44 and RHAMM (receptor for HA-mediated motility) are known as the two main HA receptors [7]. Besides serving as a structural element and an extracellular reservoir to hold large amounts of water and metal ions to maintain tissue homeostasis, high molecular weight-HA (HMW-HA) possesses antiangiogenic and antiproliferative properties. HMW-HA binding with CD44 mediates CD44 clustering, resulting in the activation of the tumor-suppressive Hippo pathway [8]. In contrast, low molecular weight-HA (LMW-HA) possesses pro-tumor functions. LMW-HA can non-covalently interact with different receptors (e.g. CD44, RHAMM and TLRs), mediating signaling cascades related to many cellular functions, including cell proliferation, differentiation, and migration [9].

In addition to other known cancer targets, the HA-synthesizing and -degrading enzymes, as well as HA receptors (e.g.CD44 and RHAMM) are upregulated in NSCLC cells, resulting in enhanced HA turnover through its synthesis and degradation [10]. A high level of the resulting LMW-HA corresponds with tumor progression and metastasis [11,12]. Both HA synthases and hyaluronidases are therefore emerging as promising targets for cancer treatment [13–15]. Previous reports have shown that inhibition of the interaction between LMW-HA and CD44/ RHAMM can be a potential approach for lung cancer treatment. For example, the downregulation of RHAMM could induce A549 NSCLC cell apoptosis upon radiotherapy [16]. Insulinlike growth factor binding protein-3 (IGFBP-3) was found to inhibit A549 cell viability by binding to HA and blocking HA-CD44 signaling [17]. Emodin [18] and triptolide [19] could downregulate HAS2 expression in NSCLC cells, reducing the HA production and secretion, and consequently inducing the cell-cycle arrest and inhibiting the cell proliferation. 3-Fluoro-N-(2-((1,2,3,4-tetrahydroacridin-9-yl)amino)ethyl)benzamide, a 3-fluorobenzoic acid derivative, was found to inhibit hyaluronidase activity, corresponding to its antiproliferation against A549 cells [20].

In addition to synthetic molecules, many crude extracts and naturally occurring compounds were shown to possess hyaluronidase inhibitory activity. For example, the crude hydroalcoholic extract and its butanol subfraction from the leaves of *Ravenala madagascariensis* (Sonn.) could inhibit hyaluronidase activity [21]. Bioassay-guided isolation and metabolomic analysis led to the identification of flavonoids, narcissin, rutin, and quercetin-3-O-glucoside, as promising hyaluronidase inhibitors. The extract containing rosmarinic acid, a phenylpropanoid, from *Melissa Officinalis* Linn. possessed anti-allergic activity presumably by its potent suppressive effect on hyaluronidase [22]. Five phenylpropanoids including clinopodic acid C, lycopic acid A, clinopodic acid E and lycopic acid B, isolated from *Lycopus lucidus* Turcz (Lamiaceae), showed hyaluronidase inhibitory activity comparable to that of rosmarinic acid [23]. Plants are therefore a potential source of hyaluronidase inhibitors.

*Xenostegia tridentata* (L.) D.F. Austin & Staples was recorded as a main component in *Prasaranadi Kashayam*, a well-known traditional Indian medicine [24]. As folk wisdom, this medicinal plant has been used to cure many diseases, including rheumatism, skin infections, fever, diabetes, diarrhea, and urinary disorders. Flavonoids, phenolics, ergosine alkaloids, pyrrolidine alkaloids like hygrine and nicotine have been isolated from the plant [25]. In addition to previous extensive studies on its broad biological activities, including antioxidant,

antidiabetic, anti-inflammatory, anti-arthritis, analgesic, wound healing, antimicrobial and larvicidal activities [26–29], our group recently discovered that the ethanol extract of *X. tridentata* also possesses anti-allergic activity in both *in vitro* and *in vivo* models [30]. Corresponding to the literature that *X. tridentata* is a flavonoid- and polyphenol-rich plant, 3,5-dicaffeoylquinic acid, quercetin-3-O-rhamnoside, kaempferol-3-O-rhamnoside, and luteolin-7-O-glucoside isolated from the ethyl acetate subfraction were found to be responsible for the observed anti-allergic activities. Despite extensive studies of the phytochemicals and bioactivities of *X. tridentata*, knowledge of its chemical components is still limited, and its anticancer activity has never been revealed.

Hyaluronidase is a potential anticancer target and many polyphenols and flavonoids such as luteolin, quercetin, and kaempferol, the components present in *X. tridentata*, are known to possess hyaluronidase inhibitory activity. Accordingly, the anticancer activities against A549 cells and hyaluronidase inhibition of the extracts from *X. tridentata* were evaluated in this study. Bioassay-guided isolation was also performed to identify the active hyaluronidase inhibitors derived from the plant.

## 2. Materials and methods

### 2.1. General information

All chemicals and solvents used were at least of analytical grade. Solvents were purchased from RCI Labscan Limited. Cisplatin and quercetin were purchased from TCI Japan and Sigma Aldrich, respectively. $^1$H and $^{13}$C NMR spectra were recorded at 298 K on a Bruker Avance 300 MHz or a Bruker ASCEND 600 MHz NMR spectrometer. High-resolution mass spectra (HRMS) were obtained using ESI or APCI ionization mode on an Orbitrap Fusion Tribrid mass spectrometer (Thermo, Massachusetts, USA).

### 2.2. Preparation of *X. tridentata* crude extracts

**Plant material.**  *Xenostegia tridentata* (L.) D.F. Austin & Staples was harvested in Chonburi province, Thailand 13˚05'54.2"N and 101˚09'47.2"E in January 2018. The plant species was kindly identified by Dr. Pranee Nangam at Faculty of Science, Naresuan University and the specimen was deposited at the PNU plant herbarium, Department of Biology, Faculty of Science, Naresuan University, Thailand (Voucher ID: 004662). The sample was washed with distilled water and shade-dried before grinding into powder and storing at -20˚C until used. Since the plant sample was collected from roadside area, the study did not need specific permissions or licenses.

**Preparation of crude ethanol extract.**  The powder of the air-dried aerial part (50.0 g) of *X. tridentata* was macerated with absolute ethanol (500 mL) at room temperature for 24 h. The extract was filtered, and the residue was re-extracted twice as described. The filtrates were combined and evaporated under vacuum at 40˚C to dryness by using a rotary evaporator to yield crude ethanol extract (cr. EtOH) as a green viscous liquid (5.2 g, yield: 10.4%).

**Partition of crude ethanol extract.**  The crude EtOH extract (5.0 g) was added with 50 mL of distilled water. Sequential extraction with hexane (5 x 50 mL), ethyl acetate (5 x 50 mL), and n-butanol (2 x 50 mL), respectively, was performed. Each of the resulting partitions and the remaining fraction (water) were combined separately and concentrated under vacuum at 40˚C to yield four subfractions, namely Hex (2.1 g, yield: 42%), EA (0.2 g, yield: 4%), BuOH (0.6 g, yield: 12%), and H2O (1.2 g, yield: 24%), respectively.

## 2.3. Evaluation of anticancer activities

**Cell culture.** A549 and L929 cell lines were obtained from National center for genetic engineering and biotechnology (BIOTEC) and Prof. Tanapat Palaga, department of microbiology, faculty of science, Chulalongkorn university, respectively. Cells were cultured in Dulbecco's modified Eagle's medium (DMEM; Gibco) supplemented with 10% fetal bovine serum (FBS; HyClone) and 1% penicillin-streptomycin (Gibco), under 5% $CO_2$, at 37˚C.

**MTT assay.** The cytotoxicity was determined by using 3-(4,5-Dimethylthiazol-2-yl)-2,5-Diphenyltetrazolium Bromide (MTT) assay. Briefly, the cells were seeded at 10,000 cells/well in a 96-well plate and allowed to grow for 16 h before incubation with solutions of the samples at the designated concentrations or DMSO (vehicle control) in the culture medium. After 48 h of incubation, the cells were added with MTT solution (0.5 mg/mL in culture medium) and incubated at 37˚C for 3 hours. The solution was then replaced with 100 μL of DMSO, and absorbance at 570 nm was measured using a microplate reader (Varioskan LUX multimode microplate reader, Thermo Scientific). The %inhibition was calculated relative to the vehicle control, and $IC_{50}$ values were calculated by using GraphPad Prism 5.0 software.

**Scratch assay.** The inhibition of cancer cell migration was evaluated by using a scratch assay. A549 cells were plated into a 24-well plate at a density of $2x10^5$ cells/well and allowed to form a confluent monolayer for 24 h. Subsequently, the monolayer was scratched with a sterile pipette tip (200 μL) and washed with PBS buffer to remove floating and detached cells. The cells were then treated with the sample or DMSO (vehicle control) in culture medium. The scratched areas were photographed (magnification, x4) at 0, 24, and 48 h. The degree of cell migration was quantified by measuring the percentage change in wound area at each time point relative to the initial wound area.

## 2.4. Determination of hyaluronidase inhibitory activity

The hyaluronidase inhibitory activity of the extracts and compounds derived from *X. tridentata* was determined by using a turbidimetric assay. The experiment was performed by following the protocol of Hyaluronidase Inhibitor Screening Assay Kit (Sigma Aldrich) with modification. In brief, 40 μL of hyaluronidase enzyme solution (10 U/mL, Sigma Aldrich: H3506) in enzyme buffer (20 mM sodium phosphate with 77 mM sodium chloride and 0.01% (w/v) bovine serum albumin, pH 7.0 at 37˚C) was incubated with 20 μL of the tested samples or DMSO (vehicle control) in assay buffer (300 mM sodium phosphate, pH 5.35 at 37˚C) for 15 min at room temperature. Then, 40 μL of 0.3% (w/v) hyaluronic acid (Sigma Aldrich: H5388) in the assay buffer was added, and the mixture was further incubated for 20 min at room temperature. The enzymatic reaction was then stopped by adding 160 μL of stop buffer (24 mM sodium acetate, 79 mM acetic acid with 0.1% (w/v) bovine serum albumin, pH 3.75 at 25˚C) and the mixture was incubated for 10 min at room temperature. Enzyme activity was then quantified by measuring absorbance at 600 nm using a microplate reader (Varioskan LUX multimode microplate reader, Thermo Scientific) to determine turbidity caused by the remaining of the undegraded hyaluronic substrate. The percentage inhibition was calculated according to the following equation.

$$\%\text{inhibition} = (1-((OD_{NEC}-OD_{sample})/(OD_{NEC}-OD_{NIC}))) \text{ x } 100$$

Where, $OD_{NEC}$ = OD value at 600 nm of the no enzyme control (NEC)

$OD_{NIC}$ = OD value at 600 nm of the vehicle control (NIC)

$OD_{sample}$ = OD value at 600 nm of the tested sample

## 2.5. Immunofluorescence staining for hyaluronan

A549 cells were grown and treated on a cell culture chamber slide. After 24 h of incubation, the cells were washed with PBS before fixing with 4% formaldehyde in PBS for 10 min at room temperature. The cells were permeabilized with 0.1% Triton X-100 in PBS for 5 min. After thorough washing with PBS, blocking was performed by incubation with 5% BSA in PBS for 1 h at room temperature. To determine the hyaluronan content, the cells were incubated with 2 μg/mL biotinylated- hyaluronic acid binding protein (b-HABP) in PBS at 4°C overnight. After washing with PBS, Alexa-Flour 488-Streptavidin conjugate (Invitrogen, Thermo Fisher Scientific, 2 μg/mL in 1% BSA in PBS) was added and incubated for 1 h at room temperature in the dark. DAPI (1 μg/mL) in PBS was used to stain the nuclei. The cells were washed with PBS before mounting on coverslips using MOWIOL mounting medium. The fluorescent signals were then visualized by using a fluorescent microscope (Nikon Eclipse Ti2-E). The mean fluorescent intensity was analyzed by using NIS-Elements imaging software.

## 2.6. Quantification of LMW hyaluronan

The culture medium of A549 cells treated with the samples for 24 h was collected and centrifuged through Amicon Ultra-0.5 centrifugal filters (100 kDa molecular weight cutoff) at 14000 g for 10 min. The filtrate was then diluted in wash buffer (0.05% v/v Tween-20 in PBS) for LMW-HA quantification by ELISA. Briefly, a 96-well plate (NUNC Maxisorp, Thermo Scientific, USA) was precoated with 1 μg/mL HABP in coating buffer (50 mM carbonate buffer (pH 9.6)) at 4°C overnight and blocked with 1% BSA in PBS at room temperature for 1 h. After washing with the wash buffer, 100 μL of the prepared samples were then transferred into the treated plate and incubated at 4°C overnight. After washing, the samples were incubated with 100 μL b-HABP (0.5 μg/mL) in wash buffer at room temperature for 1 h. Excess b-HABP was then removed by washing with wash buffer. The bound b-HABP was determined by incubating with 100 μL streptavidin-horse radish peroxidase (1:2000 dilution in wash buffer, Biotechne R&D system, DY998) and developing with 100 μL TMB (3, 3',5, 5'-tetramethylbenzidine, Sigma Aldrich, T4444). The reaction was terminated with 50 μL 2M $H_2SO_4$ and the absorbance was measured at 450 nm by using a microplate reader (Varioskan LUX multimode microplate reader, Thermo Scientific). The amount of LMW-HA was calculated according to the standard curve of hyaluronic acid (Sigma Aldrich: H5388).

## 2.7. Compound isolation

2.0 g of the hexane subfraction was further isolated by using silica gel (70–230 mesh, from Silicycle with catalog number R10040B) column chromatography. The column (30 x 5 cm) was sequentially eluted with EtOAc:Hexane 0:10–10:0 (v/v) and MeOH:$CH_2Cl_2$ 2:8 (v/v), respectively. The fractions obtained from the column were combined into 6 fractions (F1-6), based on thin-layer chromatography (TLC) profiles. By using silica gel column chromatography, fractions F3 and F5 were further isolated by elution with EtOAc:Hexane 0:10–1:9 (v/v) followed by EtOAc:$CH_2Cl_2$ 0:10–1:9 (v/v) for F3, and EtOAc:$CH_2Cl_2$ 0:10–2:8 (v/v) for F5, to yield compounds **1**–**6**. The chemical structures of the isolated compounds were elucidated by using NMR spectroscopy and mass spectrometry techniques and compared with previously published data. For the ethyl acetate fraction, compounds **7**–**10** were isolated according to our previous report [30]. HPLC analysis was performed to determine the purity of the isolated compounds.

## 2.8. X-ray crystallographic analysis of compound 1

A suitable crystal of compound **1** ($C_{30}H_{50}O$) was selected and mounted on a SuperNova, Single source at offset/far, HyPix3000 diffractometer. The crystal was kept at 293(2) K during data collection. Using Olex2, the structure was solved with the SHELXT structure solution program using Intrinsic Phasing and refined with the SHELXL refinement package using Least Squares minimization [31–33]. The crystal structure was refined in monoclinic, space group $C2$ (no. 5), $a = 36.695(4)$ Å, $b = 7.6196(15)$ Å, $c = 10.733(2)$ Å, $\beta = 92.017(15)°$, V = 2999.1(9) Å$^3$, Z = 4, 3706 reflections measured ($3.798° \leq 2\theta \leq 54.462°$), 2628 unique ($R_{int}$ = 0.0752, $R_{sigma}$ = 0.1630) which were used in all calculations. The final $R_1$ was 0.0897 (I > 2σ(I)) and $wR_2$ was 0.3009 (all data). The crystallographic information is listed in S1 Table.

## 2.9. Computational simulation

**Protein and ligand preparation.** The crystal structure of human hyaluronidase 1 (Hyal-1) with a resolution of 2.0 Å was obtained from the Protein Data Bank (https://www.rcsb.org; PDB ID: 2PE4) [34] and downloaded using the accession number 2PE4. To prepare the protein for docking simulation, chain A of Hyal-1 was selected, bound ligands and water molecules were removed, using BIOVIA Discovery Studio Visualizer (2021) and saved in PDB format. The protein file was opened in AutoDockTools 1.5.6 and saved in PDBQT format.

The selected ligands (compounds **1–10**, quercetin, and hyaluronan) were drawn using ChemDraw Profesional 16.0, viewed on BIOVIA Discovery Studio Visualizer (2021), and saved in PDB format. The ligand files were opened in AutoDockTools 1.5.6 and saved in PDBQT format.

**Grid box preparation.** The grid box for molecular docking was created with AutoGrid in AutoDockTools 1.5.6 using the previously reported grid box center coordinates [35] to cover all amino acid residues at the active site [34]. The grid box dimensions were specified at 80 Å for the x, y, and z axes, respectively. The grid box center coordinates were specified at 37.045, -17.292, and -11.844 for the x, y, and z axes, respectively [35].

**Molecular docking.** Computational simulation studies were conducted using AutoDock4 [36]. The configuration file was prepared using the aforementioned grid box center coordinates and dimensions. The simulation was performed for 1,000 runs for each compound. Once the docking simulations were completed, the results were visualized and analyzed using BIOVIA Discovery Studio Visualizer (2021).

## 2.10. Statistical analysis

GraphPad Prism 5.0 software was used for statistical analysis. One-way ANOVA and Dunnett's multiple comparison test were used to evaluate the statistical difference. $p < 0.05$ was considered to indicate a significant difference.

## 3. Results and discussion

### 3.1. Anticancer effects of *X. tridentata* extracts against A549 cells

In this study, ethanol was used as the extracting solvent due to its polarity and ability to efficiently dissolve a wide range of bioactive compounds. The crude ethanol extract of *X. tridentata* and its subfractions were evaluated for the cytotoxicity against A549 cells. As shown in **Fig 1A**, from MTT assay, the crude ethanol extract inhibited A549 cancer cell growth in a dose-dependent manner, albeit with weak potency. Among subfractions, hexane extract exhibited the strongest cytotoxicity, with significantly higher activity than that of the crude ethanol extract and other subfractions at 250 μg/mL. To examine selectivity of the active extracts

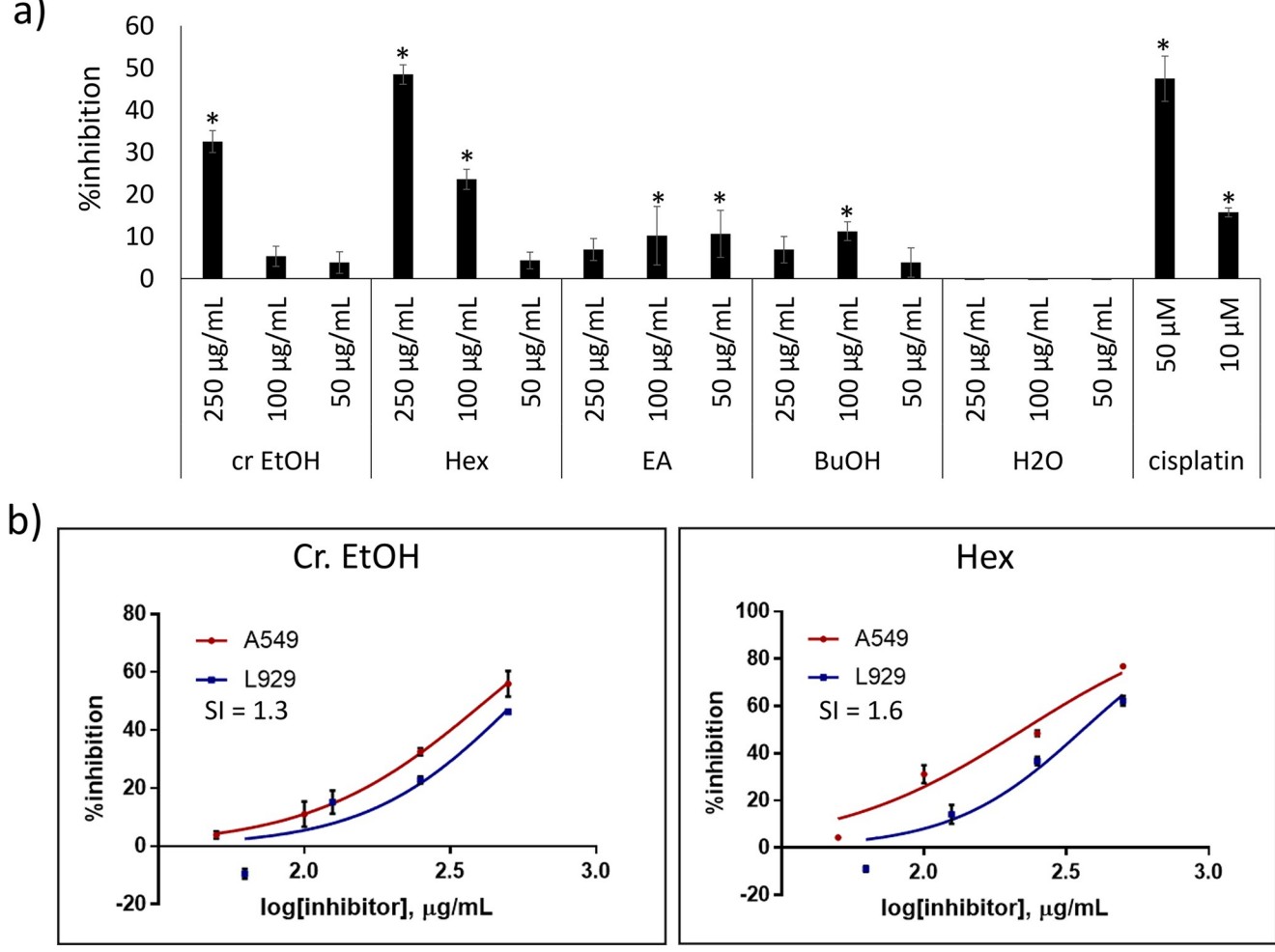

**Fig 1. Cytotoxicity of crude EtOH extract of *X. tridentata* and its subfractions.** a) A549 cells were treated with the crude EtOH extract, subfractions or cisplatin (reference compound) at the designated concentrations for 48 h. The cell viability was determined by using MTT. The percentage of inhibition was calculated based on the absorbance at 570 nm, relative to the absorbance of the vehicle control. *$p < 0.05$, compared to the vehicle control. b) Selective toxicity of crude EtOH extract and Hex subfraction against cancer cells. The selectivity index (SI) was calculated as the ratio of the $IC_{50}$ value against L929 normal cells to the $IC_{50}$ value against A549 cancer cells.

toward cancer cells, the MTT assay was also performed on L929 normal cells. The result shown in **Fig 1B** indicated that both the crude ethanol extract and the hexane subfraction were more toxic to cancer cells than to normal cells with selectivity indexes of 1.3 and 1.6, respectively. Interestingly, although the ethyl acetate subfraction possessed only a weak cytotoxic effect, it showed the significant inhibitory activity on A549 cell migration at both 24 and 48 hours of incubation, and its activity was higher than that of the ethanol extract (**Fig 2**). Although the crude ethanol extract of *X. tridentata* exhibited only moderate anticancer activity, findings from the aforementioned study indicated that further refinement enhanced its efficacy, warranting additional investigation into its active components and mode of action.

### 3.2. Hyaluronidase inhibitory activity of *X. tridentata* extracts

To evaluate the inhibition of hyaluronidase activity, a turbidity assay was performed. The result shown in **Table 1** indicated that the hyaluronidase inhibitory activity was not observed

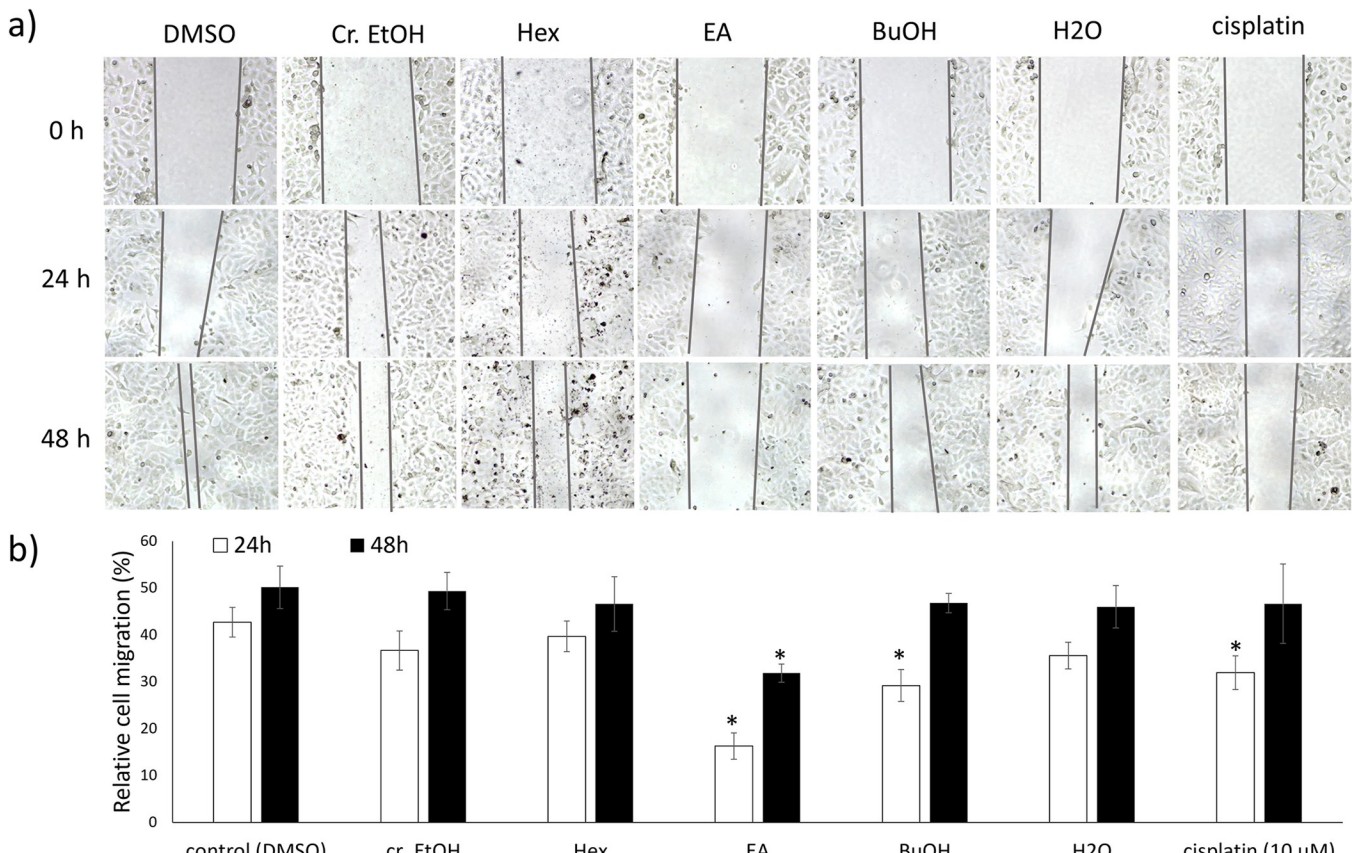

**Fig 2. Inhibitory activity of the crude EtOH extract of *X. tridentata* and its subfractions on A549 cell migration.** a) A549 cells were plated into a 24-well plate at a density of 2x10⁵ cells/well and allowed to form a confluent monolayer for 24 h. The monolayer was scratched and washed with PBS buffer to remove floating and detached cells. The cells were then treated with the extracts (50 μg/mL), cisplatin (10 μM) or DMSO (vehicle control) in culture medium. The scratched areas were photographed (magnification, x4) at 0, 24, and 48 h. b) The extent of cell migration was evaluated by measuring the percentage change in wound area at each time point relative to the initial wound area. The data are presented as mean ± SD (n = 3). $^{*}p < 0.05$, compared to the control.

**Table 1. Hyaluronidase inhibitory activity of crude ethanol extract of *X. tridentata* and its subfractions from turbidity assay.**

| sample | concentration | % inhibition | sample | concentration | % inhibition |
|---|---|---|---|---|---|
| Cr. EtOH | 200 μg/mL | NI | EtOH/BuOH | 200 μg/mL | 12.0 ± 4.3* |
| | 100 μg/mL | NI | | 100 μg/mL | NI |
| | 50 μg/mL | NI | | 50 μg/mL | NI |
| EtOH/Hex | 200 μg/mL | 10.2 ± 2.8* | EtOH/H2O | 200 μg/mL | 4.2 ± 1.3 |
| | 100 μg/mL | 2.7 ± 0.7 | | 100 μg/mL | 1.8 ± 3.1 |
| | 50 μg/mL | 1.6 ± 1.3 | | 50 μg/mL | 0.3 ± 2.7 |
| EtOH/EtOAc | 200 μg/mL | 83.7 ± 1.7* | Quercetin | 30.2 μg/mL | 32.0 ± 1.1* |
| | 100 μg/mL | 54.7 ± 2.3* | | 15.1 μg/mL | 17.4 ± 2.5* |
| | 50 μg/mL | 25.2 ± 1.5* | | | |

NI: The inhibitory activity was not observed.

The data are presented as mean ± SD (n = 4).

$^{*}p < 0.05$, compared to the control.

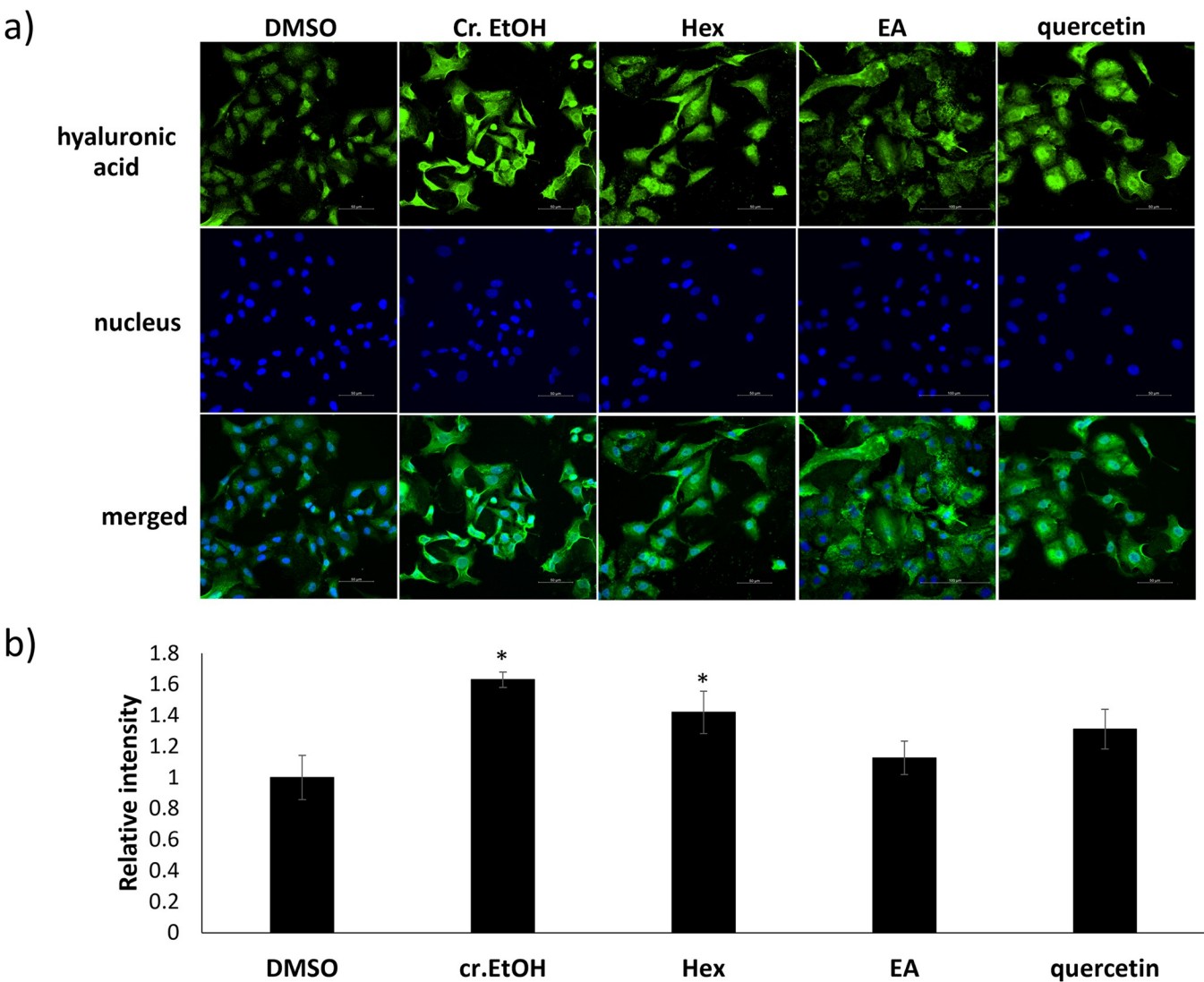

**Fig 3. Immunofluorescence staining for hyaluronan.** a) Representative images of A549 cells treated with *X. tridentata* extracts (200 μg/mL), quercetin (100 μM) or DMSO (vehicle control) for 24 h before staining with biotinylated-HABP (b-HABP) and Alexa-Flour 488-Streptavidin conjugate (green) and DAPI (blue). The images were captured with 40x objective and integrated 1.5x tube lens. b) Quantification of the mean fluorescent intensity (green) of A549 cells treated with the samples as compared to that of DMSO. The data are presented as mean ± SD. *$p < 0.05$, compared to the control.

for the crude ethanol extract of *X. tridentata* at the maximum tested concentration of 200 μg/mL. However, its ethyl acetate and hexane subfractions, respectively, exhibited good and moderate dose-dependent inhibition in the same manner as quercetin, a known hyaluronidase inhibitor. To evaluate the inhibition at the cellular level, hyaluronan of A549 cells was visualized and quantified by using immunofluorescence staining. As shown in **Fig 3**, A549 cells treated with the crude ethanol extract, the hexane and ethyl acetate subfractions, and quercetin showed a cable-like structure (green fluorescence signal), suggesting a higher amount of HMW-HA, as compared with the vehicle control experiment. Interestingly, the trend was not consistent with that of the turbidity assay where the ethyl acetate fraction showed the best inhibitory activity, presumably due to the different sources of hyaluronidases used. The enzyme used in the turbidity assay was obtained from Sigma-Aldrich (H3506) and it contained mainly Hyal-1 and Hyal-2. On the other hand, the most overexpressed hyaluronidase found in

A549 cells is TMEM2 [37,38]. Nonetheless, the hyaluronidase inhibitory activity of the crude ethanol extract and the hexane and ethyl acetate subfractions correlated with the aforementioned anticancer effects. The hexane and ethyl acetate fractions were consequently further isolated to identify potential hyaluronidase inhibitors derived from *X. tridentata*.

### 3.3. Isolation of hyaluronidase inhibitors from *X. tridentata* extracts

The NMR spectrum of the hexane subfraction suggested the presence of a complex mixture of nonpolar components, including glycerides and fatty acids. Nonetheless, upon isolation by using silica gel column chromatography technique, six known compounds including fernenol (**1**), methyl-3,4-seco-8*β*H-fernadienoate (**2**), 2(4'-hydroxyphenyl)-ethyl behenate (**3**), 3,4-seco-8*β*H-fernadienoaic acid (**4**), lutein (**5**) and glyceryl palmitate (**6**) were isolated, as depicted in **Fig 4**. Their chemical structures were elucidated by using NMR and HRMS techniques and compared with previously reported spectra (see S1 and S2 Figs). The structure of fernenol (**1**) was also confirmed by using single-crystal X-ray diffraction (**Fig 4**). For the ethyl acetate subfraction, the phenolic and flavonoid glycoside compounds **7**–**10** were obtained, according to our previous report [30]. The chemical structures are shown in **Fig 5**. The HPLC chromatograms of the isolated compounds are shown in S2 Fig.

### 3.4. Hyaluronidase inhibitory activity of the compounds isolated from *X. tridentata*

From the turbidity assay (**Table 2**), compounds **1**–**10** isolated from the hexane and ethyl acetate subfractions showed dose-dependent inhibition of hyaluronidase activity at different potencies. According to the literature, many naturally occurring compounds, including phenolic compounds, flavonoids, triterpenes, and fatty acids have been reported to possess hyaluronidase inhibitory activity [10,39]. Correspondingly, our results showed that the phenolic and flavonoid compounds (**7**–**10**), isolated from the ethyl acetate subfraction, exhibited moderate inhibitory activity. 3,5-dicaffeoylquinic acid (**8**) possessed higher inhibitory activity than the isolated flavonoid glycosides. Quercetin-3-O-rhamnoside (**9**), exhibited weaker activity than its corresponding aglycone, quercetin [40]. The compounds isolated from the hexane subfraction also exhibited the inhibition in a dose-dependent manner, however with weaker activity than that of phenolic and flavonoid compounds, except lutein (**5**). Although numerous biological activities of these compounds have been previously reported, the information on their anticancer effects is still limited, and their hyaluronidase inhibition has never been revealed. Fernenol (**1**), a pentacyclic triterpene, was reported as a component in many plant extracts, for example, *Citrullus Colocynthis L*. and *Artemisia Vulgaris L* [41–43]. According to the literature search, only its antifungal activity has been reported [44]. The only report about 2(4'-hydroxyphenyl)-ethyl behenate (**3**) was its isolation from *Buddleja cordata* subsp. *cordata* and its moderated antituberculosis activity [45]. The tetracyclic triterpenes, 3,4-seco-8*β*H-fernadienoic acid (**4**) and its methyl ester (**2**), were isolated from *Euphoebia Chamaesyce* [46]. Interestingly, compound **4** was found to possess stronger activity than its ester derivative, compound **2**, on the antiproliferation against human cancer cell lines, including A549 [47], and DNA topoisomerase inhibition [48].

Since compound **5**, among all tested compounds, showed to be the most potent hyaluronidase inhibitor in our *in vitro* assay, the cell-based hyaluronidase inhibitory activity of compound **5** was then visualized by immunofluorescence staining. As shown in **Fig 6A and 6B**, the green fluorescence intensity of A549 cells treated with compound **5** was dose-dependently increased, indicating higher hyaluronidase inhibition, as compared to the vehicle control experiment. In addition, ELISA was performed to determine the amount of LMW HA. As

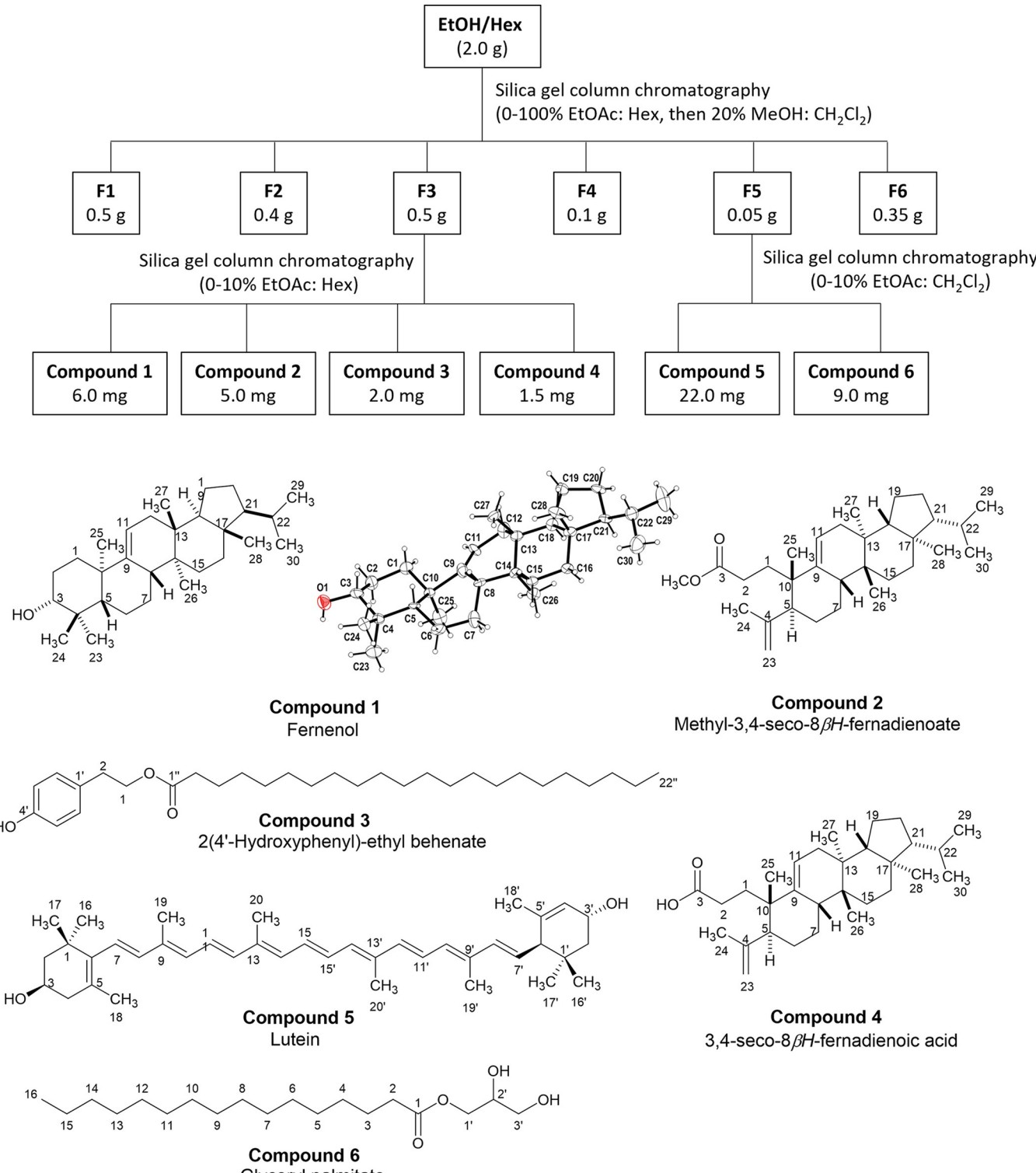

**Fig 4. Isolation of hexane subfraction and chemical structures of the isolated compounds.** The asymmetric unit of compound **1** is shown.

**Fig 5. Isolated compounds from ethyl acetate subfraction.**

shown in **Fig 6C**, the levels of LMW HA in the culture medium of A549 cells treated with crude ethanol, its hexane subfraction, and compound **5** were significantly lower than that of vehicle control, assuring their ability to inhibit the HA degradation. Notably, this is the first

**Table 2. Hyaluronidase inhibitory activity of compounds isolated from hexane and ethyl acetate subfractions from turbidity assay.**

| conc. (μM) | % inhibition | | | | | |
|---|---|---|---|---|---|---|
| | Compound 1 | Compound 2 | Compound 3 | Compound 4 | Compound 5 | Quercetin |
| 100 | 10.7 ± 0.7 | 10.2 ± 0.7 | 15.3 ± 4.2 | 10.2 ± 0.9 | 51.5 ± 9.6* | 32.0 ± 1.1* |
| 50 | 3.9 ± 6.9 | 9.0 ± 0.6 | 10.6 ± 3.0 | 6.6 ± 1.8 | 20.6 ± 1.3* | 17.4 ± 2.5* |
| 25 | 2.6 ± 6.3 | 7.2 ± 1.2 | 1.9 ± 2.9 | 3.9 ± 1.9 | 10.8 ± 1.8 | 13.2 ± 2.6 |
| 12.5 | NI | NI | NI | 2.6 ± 2.2 | 2.7 ± 3.0 | 5.0 ± 8.1 |
| conc. (μM) | Compound 6 | Compound 7 | Compound 8 | Compound 9 | Compound 10 | |
| 100 | 5.4 ± 3.1 | 14.9 ± 5.0 | 31.3 ± 0.6* | 18.8 ± 3.8 | 17.2 ± 2.5 | |
| 50 | 1.6 ± 1.1 | 10.7 ± 3.2 | 17.9 ± 2.0* | 18.9 ± 3.8* | 14.7 ± 1.5 | |
| 25 | NI | 8.9 ± 3.8 | 8.8 ± 3.4 | 9.4 ± 3.2 | 10.5 ± 2.6 | |
| 12.5 | NI | 3.8 ± 5.8 | 4.2 ± 4.9 | 3.7 ± 8.5 | 1.6 ± 7.4 | |

NI: The inhibitory activity was not observed.

The data are presented as mean ± SEM, n = 3.

*$p < 0.05$, compared to the control.

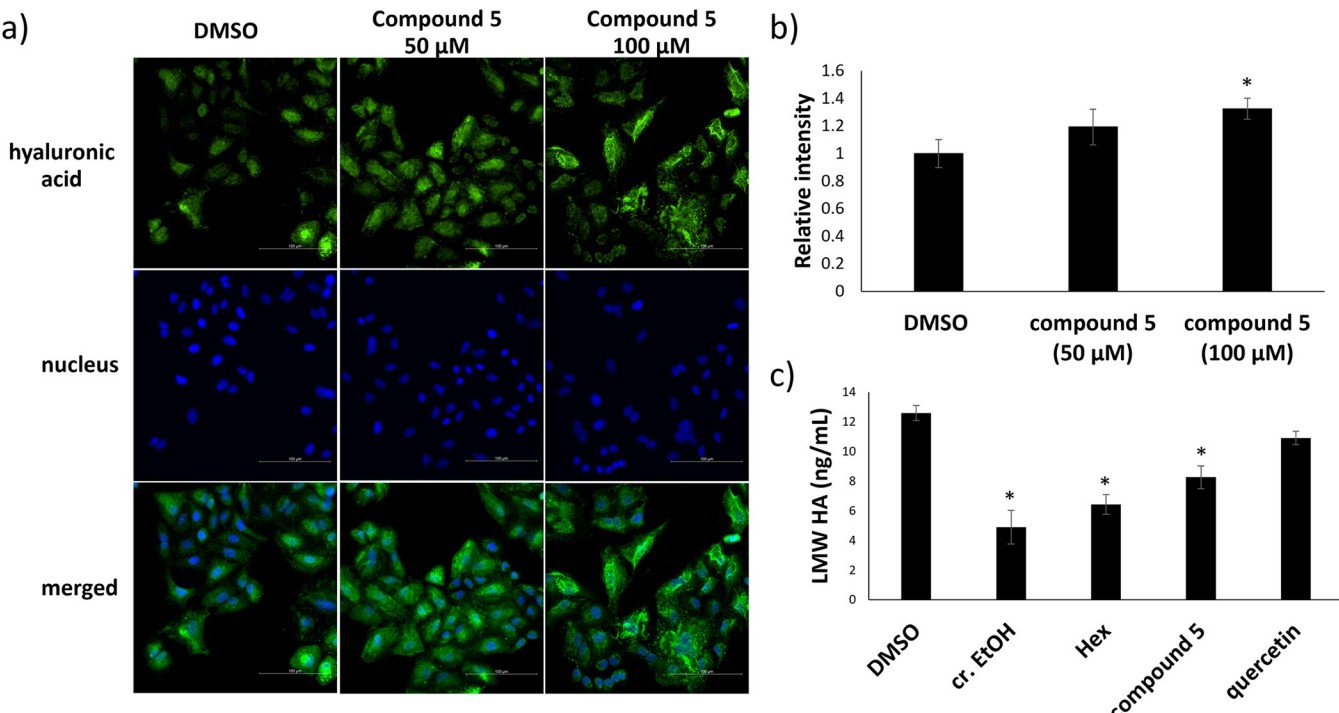

**Fig 6. Cell-based hyaluronidase inhibitory activity.** a) Representative images of A549 cells treated with compound **5** (50 and 100 μM) or DMSO (vehicle control) for 24 h before staining with biotinylated-HABP (b-HABP) and Alexa-Flour 488-Streptavidin conjugate (green) and DAPI (blue). The images were captured with 40x objective and integrated 1.5x tube lens. b) Quantification of the mean fluorescent intensity (green) of A549 cells treated with the samples as compared to that of DMSO. The data are presented as mean ± SD. *$p<0.05$, as compared to the control. c) ELISA quantification of low molecular weight (LMW) hyaluronic acid in the culture medium of A549 cells treated with the extract of *X. tridentata* (200 μM), compound 5 (100 μM) or DMSO (vehicle control) for 24 h. The data are presented as mean ± SD. *$p < 0.05$, compared to the control.

report on the ability of lutein on hyaluronidase inhibition despite many studies on its potential as an anticancer agent against A549 cells [49–52].

To understand the molecular interactions crucial for hyaluronidase inhibition, molecular docking studies of all of the isolated compounds (compounds **1–10**) and hyaluronan were performed in comparison with quercetin [53,54] to analyze their binding interaction with Hyal-1, using the protocols as described in the 'Materials and Methods' section. The binding energies and amino acid interactions of the aforementioned molecules in Hyal-1 are summarized in **Table 3**.

As shown in **Table 3**, the computational simulation results of compounds with highly favorable binding energies (compounds **5, 8**, and quercetin) correlated well with the results from the inhibition assay (**Table 2**). Although compounds **1, 2,** and **4** also showed favorable *in silico* binding energies, they exhibit relatively low *in vitro* inhibitory activity most likely due to their poor solubilities in the assay buffer, based on our observation. On the other hand, the docking of hyaluronan showed the expected relatively unfavorable binding energy, but the result may not accurately represent the actual binding pose of the hyaluronan with much longer chain lengths in Hyal-1 in nature. The rest of the studied compounds exhibit the *in silico* binding energies that roughly correlated with the trend observed in the *in vitro* inhibition assay. Since compound **5** (Lutein) showed the distinct efficacious *in vitro* inhibitory (**Table 2**) activity and *in silico* binding energy (-11.70 kcal/mol) compared to other compounds in this study, its binding interactions with Hyal-1 were further analyzed, in comparison with quercetin and other compounds.

**Table 3. Summary of *in silico* binding energy, inhibition constant ($K_i$), and amino acid residue interactions for compounds 1–10, quercetin, and hyaluronan.**

| Compound | Lowest binding energy [kcal/mol] with estimated inhibition constant, $K_i$ | Number of interacting amino acids (AA) and interacting amino acid residues with interacting distance (Å) | Number of interactions (Number of Hydrogen-bond Interactions) |
|---|---|---|---|
| 1 | -10.9 ($K_i$ = 10.3 nM) | 3 AA = TYR84 (4.03), *ASP129* (3.31, 3.19), ILE146 (4.41) | 4 (2) |
| 2 | -8.40 ($K_i$ = 696.36 nM) | 5 AA = **GLU131** (3.33), **TYR202** (3.77, 4.68), **TYR286** (4.59), **TRP321** (4.38, 5.52), TRP324 (4.35, 4.52, 4.78) | 9 (0) |
| 3 | -5.69 ($K_i$ = 66920 nM) | 6 AA = *PRO62* (3.37), **TYR75** (3.93), *SER76* (3.28), TYR84 (4.53), ALA132 (4.01, 4.55, 4.89), ILE146 (3.75, 4.64, 4.82) | 10 (2) |
| 4 | -8.40 ($K_i$ = 696.58 nM) | 7 AA = **ARG134** (4.12), *GLY203* (3.26), PHE204 (3.85), *ASP206* (2.95), ***TYR208*** (2.67), **TYR210** (4.94), PRO249 (3.84) | 7 (3) |
| 5 | -11.7 ($K_i$ = 2.67 nM) | 6 AA = ***TYR75*** (2.08), **TYR202** (4.49), ***TYR210*** (1.92, 4.43), PRO249 (4.68), **TYR286** (3.90), **TRP321** (4.40, 4.87) | 8 (2) |
| 6 | -5.29 ($K_i$ = 132130 nM) | 9 AA = ILE73 (3.44), **GLU131** (3.60), **ARG134** (3.57), ***TYR202*** (2.68, 3.18, 4.74), GLY203 (3.31, 3.66), SER245 (3.00), **TYR247** (4.89, 4.93), **TYR286** (4.79), **TRP321** (4.74, 4.74, 4.74) | 15 (1) |
| 7 | -7.99 ($K_i$ = 1400 nM) | 10 AA = *ASN37* (2.74), **GLU131**\* (3.77, 3.91), ***ARG134*** (3.36), **TYR202** (3.42), *ASP206* (2.60, 2.86), ***TYR210*** (2.85), *SER245* (3.10), ***ARG265*** (2.57), ***TYR286*** (2.75, 4.96), **TRP321** (2.79, 4.75) | 14 (8) |
| 8 | -9.10 ($K_i$ = 135.3 nM) | 10 AA = *ASN37* (3.31), **ASP129**\* (4.15), *TRP130* (3.02), **ARG134**\* (4.93), ***TYR202*** (2.56, 2.75), PHE204 (3.34), ***TYR210*** (2.56), *SER245* (2.86), *ASP292* (2.89), **TRP321** (4.15, 4.51) | 12 (7) |
| 9 | -7.41 ($K_i$ = 3690 nM) | 11 AA = **ARG134**\* (4.26, 4.98), ***TYR202*** (2.84, 3.37), PHE204 (3.77, 4.85), ASP206\* (4.56), ***TYR208*** (3.14), **TYR210** (2.99), *SER245* (2.99), ***TYR247*** (3.16, 3.23, 3.24), *TYR261* (2.76, 2.97), *ASP292* (2.76), **TRP321** (4.87) | 17 (10) |
| 10 | -7.14 ($K_i$ = 5790 nM) | 8 AA = *PRO62* (2.89), ILE73 (4.64), TYR75 (3.95), **ASP129**\* (4.46), **GLU131**\* (3.81, 4.03), ***TYR202*** (3.22), ***TYR286*** (2.70), **TRP321** (4.72, 5.13) | 10 (3) |
| Quercetin | -8.19 ($K_i$ = 987.03 nM) | 5 AA = ILE73 (4.31), **ASP129**\* (4.39), **GLU131**\* (3.97, 4.07), **TYR286** (3.69), **TRP321** (4.45, 5.40) | 7 (0) |
| Hyaluronan | -2.24 ($K_i$ = 22650000 nM) | 10 AA = ***TYR75*** (2.60), ***ASP129*** (2.86, 4.22), **GLU131** (2.88, 2.69, 3.13, 3.66, 4.22, 5.60), *TRP141* (2.80), *ASP142* (3.00, 3.36), *THR143* (3.30), LYS144\* (2.94), **TYR202**\* (4.88), **TRP321**\* (4.76), TRP324 (3.52) | 17 (6) |

Note: Underlined residue = amino acid residues that interact with 8 or more compounds in this study; bolded residue = important amino acid residues in the active site [34]; italicized residue = residues that form hydrogen-bond interaction with the molecule; stared residue = residues that form ion interaction with the molecule.

Among all of the studied compounds, compound **5** (Lutein) was the only compound that binds across the active site of Hyal-1 (from the Western region, across the center, to the Eastern region of the binding site, **Fig 7A** and **7B**) [34], while the others either bind slightly above the active site (compounds **1** and **3**; see **S3 Fig**), at the center only (compounds **4** and **9**; see **S3 Fig**), at the Eastern region only (compounds **2**, **10**, and quercetin; see **S3 Fig** and **Fig 7A and 7D**), or around the center and the Eastern region (compounds **6**, **7**, **8**, and hyaluronan; see **S3 Fig**). This optimal binding pose also made compound **5** the only compound in this study that interacts with both TYR75 and TYR210 (two of the residues on each end of the catalytic cleft responsible for the binding of sugar molecules) [34] via hydrogen-bond interactions by using

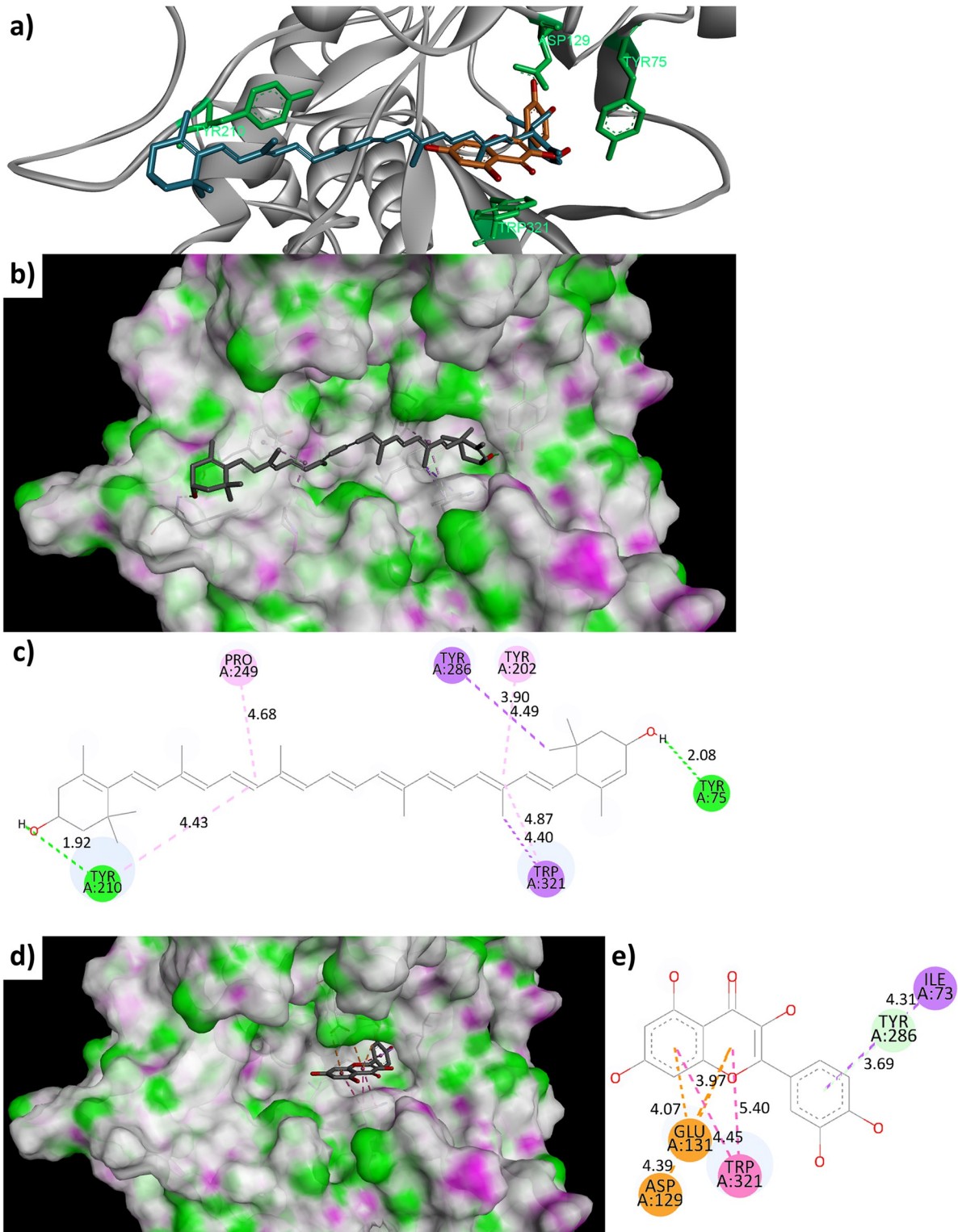

**Fig 7.** a) Docking poses of quercetin (orange) and lutein (**5**) in Hyal-1 (cyan, PDB ID: 2PE4) with four of the important amino acid residues highlighted in green; 3D representations of lutein b) and quercetin d) in the binding cleft with highlighted hydrogen-bond donor (purple) and acceptor (green) surface; 2D representation of amino acid interactions with c) lutein and e) quercetin. Note: Orange = π-anion interactions, pink = π-π interaction, light pink = π-alkyl/alky interaction, purple = π-σ interaction, green = conventional hydrogen bond interaction, and light green = π-donor hydrogen bond interaction.

the 3' and 3 hydroxyl groups on both ends of the molecule (**Fig 7C**). Furthermore, this binding pose allows compound **5** to interact with TYR202, TYR286, and TRP321 (**Fig 7A-7C**), which is located on the catalytic cleft of the active site of Hyal-1 [34], via hydrophobic interactions. On the other hand, in combination with other interactions, the interactions of quercetin with the important amino residues ASP129 and GLU131 (involved in the catalytic cleavage of the β (1,4)-linkages) [55] via π-anion interactions (**Fig 7E**) most likely explain the good *in silico* binding energy and the observed *in vitro* biological activities. Nevertheless, lutein (with 2 hydrogen bonds and 6 nonbonded interactions) exhibits significantly better *in silico* binding energy (**Table 3**) and better *in vitro* biological activities (**Table 2**) than both quercetin (with 0 hydrogen bonds and 7 nonbonded interactions) and compound **8** (with 7 hydrogen bonds and 5 nonbonded interactions). Based on our docking simulation, these observations are plausibly the result of the hydrogen-bond interaction of lutein with TYR75 (Eastern region) and TYR210 (Western region) combined with its multiple hydrophobic interactions with amino acid residues (TYR202, TYR210, TYR286, and TYP321) along the aromatic-residue-enriched sugar binding cleft [34] (**Fig 7A-7C**), which is important for Hyal-1 activity [55]. These interactions ultimately contribute to the observed potency of this particular molecule.

### 3.5. Anticancer activities of compound 5 (lutein)

Lutein is a tetraterpenoid present in fruits, vegetables, and egg yolks. Previous studies have revealed that lutein possesses promising chemopreventive effects and anticancer activity against various types of cancers, including lung cancer [49–52]. Herein, the anticancer activities of compound **5** (lutein) isolated from *X. tridentata* were validated and compared with those of compounds **1** and **2**. The result correlated with the literature [49–52] indicating that compound **5** could dose-dependently inhibit A549 NSCLC cell growth with moderate potency (**Fig 8A**). From the scratch assay, compound **5**, among the tested compounds, showed significant inhibition of cancer cell migration as compared to the control (**Fig 8B and 8C**), corresponding to its hyaluronidase inhibitory activity. A previous study revealed that lutein inhibited A549 cell proliferation by inducing DNA damage and consequently $G_0/G_1$ cell cycle arrest and apoptosis [50]. Mechanistically, Zhang W.L., et al. reported that lutein-induced A549 cell apoptosis through the inhibition of PI3K/Akt pathway, an intracellular signaling cascade related to cancer cell proliferation and migration [51]. Since many hyaluronidase inhibitors have been reported to inhibit cancer cell growth and migration by inducing intracellular ROS generation [56] and targeting the PI3K/Akt pathway [57–59], as an addition to the reported mechanism, our study suggested that compound **5** exerted the previously mentioned anticancer effects through the inhibition of hyaluronidase activity. Interestingly, despite the weak hyaluronidase inhibitory activity of compound **1**, it exhibited high potency in inhibiting A549 cell proliferation, presumably through other targets.

### 4. Conclusions

In summary, the ethanol extract of *X. tridentata* exerted anticancer activities against A549 NSCLC cells. Among the subfractions, the hexane extract exhibited the most potent antiproliferation and the ethyl acetate extract could significantly inhibit the cancer cell migration. The results corresponded to their hyaluronidase inhibitory activity. Compound isolation of the hexane and ethyl acetate fractions yielded compounds **1–10**. Among them, lutein (**5**) was found to be the most potent hyaluronidase inhibitor. The docking study suggested that lutein could bind to the binding site of Hyal-1 more efficiently than quercetin, according to the additional hydrogen-bond and hydrophobic interactions between lutein and the amino acid residues of the aromatic-residue-enriched sugar binding cleft, which is important for Hyal-1

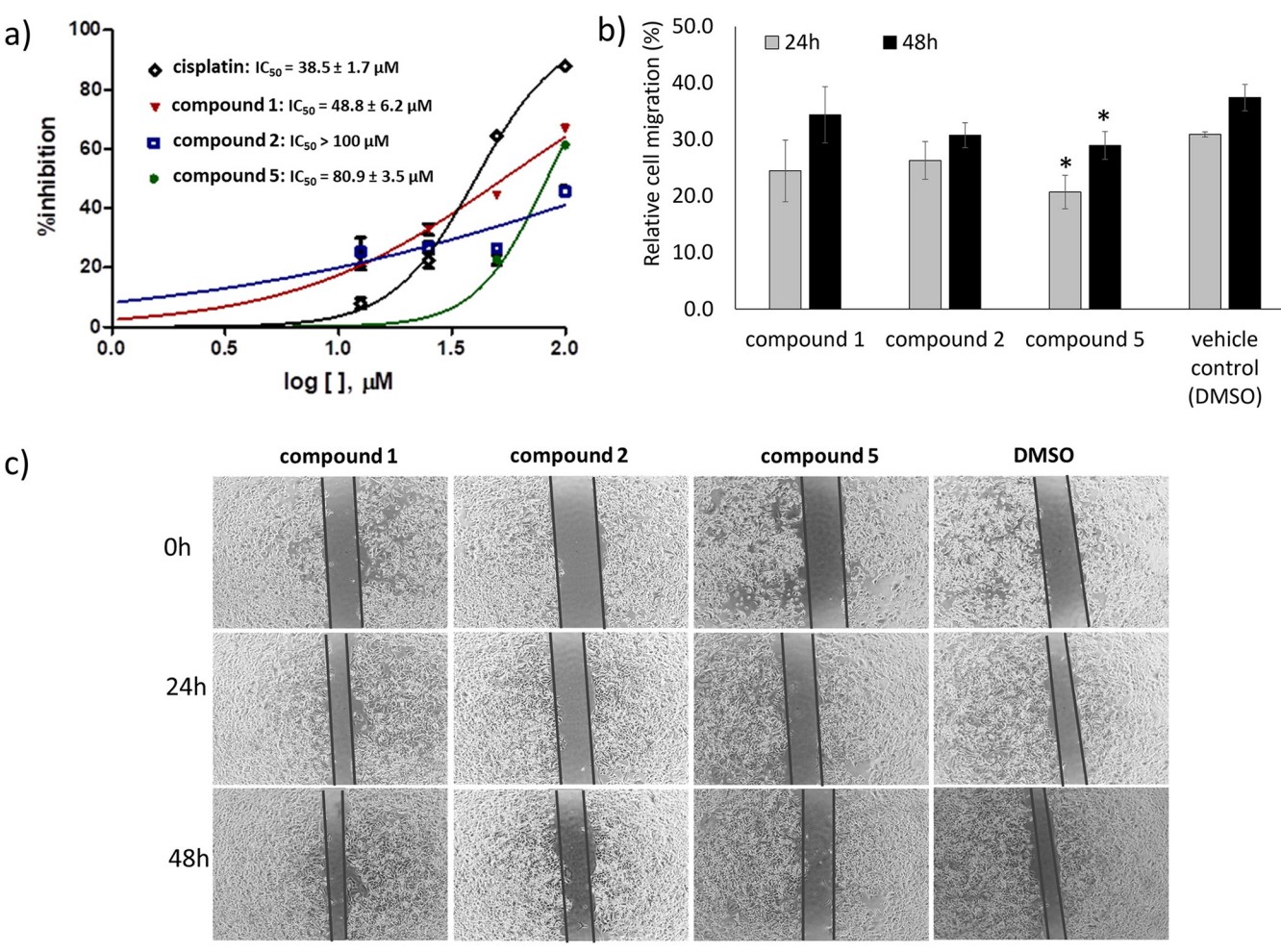

**Fig 8. Anticancer activities of the isolated compounds derived from *X. tridentata*.** a) Dose-response curve of compounds **1**, **2**, **5**, and cisplatin. A549 cells were treated with different concentrations of the compounds for 48 h. The cell viability was determined by using MTT. The percentage of inhibition was calculated based on the absorbance at 570 nm, relative to the absorbance of the vehicle control. b) Inhibition of cell migration. A549 cells were seeded in a 24-well plate and incubated for 24 hours to form a confluent monolayer. The monolayer was scratched and rinsed with PBS buffer to remove any floating or detached cells. Cells were then treated with compounds **1**, **2**, **5** (20 μM) or DMSO (vehicle control) in culture medium. Images of the scratched areas were captured (4x magnification) at 0, 24, and 48 hours. The level of cell migration was quantified by measuring the percentage change in wound area at each time point relative to the initial wound area. The data are presented as mean ± SD. $^*p < 0.05$, as compared to the control. c) The wound captured at different time points (0, 24, and 48 h).

activity. Lutein has been known to exert promising broad bioactivities, including anti-lung cancer. This work revealed that hyaluronidase could be a target of lutein, affecting PI3K/Akt pathway, the signaling cascade involved in a reported anticancer mechanism of lutein. The more in-depth study on the cellular mechanism of the *X. tridentata* extract and lutein on their anticancer activities is underway in our laboratory.

## Supporting information

**S1 Table. Crystallographic data of compound 1.**
(PDF)

**S1 Fig. Spectroscopic data of the isolated compounds.**
(PDF)

**S2 Fig. HPLC chromatograms of the isolated compounds from ethyl acetate subfraction.**
(PDF)

**S3 Fig. Docking simulation visualization.**
(PDF)

## Acknowledgments

The authors would like to thank Dr. Sutthida Wongsuwan for her assistance with the cell culture experiments and Miss Praneet Paiboonsombat for her support in project administration.

## Author Contributions

**Conceptualization:** Jaruwan Chatwichien.

**Funding acquisition:** Jaruwan Chatwichien.

**Investigation:** Jaruwan Chatwichien, Natthawat Semakul, Saranphong Yimklan, Nutchapong Suwanwong, Prakansi Naksing.

**Methodology:** Jaruwan Chatwichien, Natthawat Semakul, Saranphong Yimklan, Nutchapong Suwanwong.

**Project administration:** Jaruwan Chatwichien.

**Resources:** Jaruwan Chatwichien.

**Supervision:** Somsak Ruchirawat.

**Validation:** Jaruwan Chatwichien, Natthawat Semakul, Saranphong Yimklan, Nutchapong Suwanwong.

**Writing – original draft:** Jaruwan Chatwichien, Natthawat Semakul, Saranphong Yimklan, Nutchapong Suwanwong.

**Writing – review & editing:** Jaruwan Chatwichien.

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
