## [Decision Letter · Decision Letter 0]

30 Oct 2024

PONE-D-24-42468

Lutein derived from Xenostegia tridentata exhibits anticancer activities against A549 lung cancer cells via hyaluronidase inhibition

PLOS ONE

Dear Dr. Chatwichien,

Thank you for submitting your manuscript to PLOS ONE. After careful consideration, we feel that it has merit but does not fully meet PLOS ONE’s publication criteria as it currently stands. Therefore, we invite you to submit a revised version of the manuscript that addresses the points raised during the review process.

**ACADEMIC EDITOR: **Based on the reviewer’s evaluation, I recommend a major revision. Please try to respond thoroughly and make substantial improvements

We look forward to receiving your revised manuscript.

Kind regards,

Vinh Le Ba, PhD in Pharmaceutical Science

Academic Editor

PLOS ONE

2. We note that this submission includes NMR spectroscopy data. We would recommend that you include the following information in your methods section or as Supporting Information files:

1) The make/source of the NMR instrument used in your study, as well as the magnetic field strength. For each individual experiment, please also list: the nucleus being measured; the sample concentration; the solvent in which the sample is dissolved and if solvent signal suppression was used; the reference standard and the temperature.

2) A list of the chemical shifts for all compounds characterised by NMR spectroscopy, specifying, where relevant: the chemical shift (δ), the multiplicity and the coupling constants (in Hz), for the appropriate nuclei used for assignment.

3)The full integrated NMR spectrum, clearly labelled with the compound name and chemical structure.

We also strongly encourage authors to provide primary NMR data files, in particular for new compounds which have not been characterised in the existing literature. Authors should provide the acquisition data, FID files and processing parameters for each experiment, clearly labelled with the compound name and identifier, as well as a structure file for each provided dataset. See our list of recommended repositories here: https://journals.plos.org/plosone/s/recommended-repositories"

Please check for appropriate authentication and characterization of the samples and products used in this study, such as chemical analysis, DNA analysis, or microscopy.

“JC: Office of the Permanent Secretary, Ministry of Higher Education, Science, Research and Innovation (OPS-MHESI) [grant number RGNS 64-239] and Chulabhorn Graduate Institute

JC: Chulabhorn Royal Academy (Fundamental Fund: fiscal year 2024 by National Science Research and Innovation Fund [FRB670024/0240 Project code 198480])

NS: the Postdoctoral Research Fund from Chulabhorn Graduate Institute [grant number CGIP(2022)/ 01]”

Additional Editor Comments:

Major revision

Reviewers' comments:

Reviewer's Responses to Questions

**Comments to the Author**

1. Is the manuscript technically sound, and do the data support the conclusions?

Reviewer #1: Yes

Reviewer #2: Yes

Reviewer #3: No

2. Has the statistical analysis been performed appropriately and rigorously? 

Reviewer #1: No

Reviewer #2: Yes

Reviewer #3: I Don't Know

3. Have the authors made all data underlying the findings in their manuscript fully available?

Reviewer #1: Yes

Reviewer #2: Yes

Reviewer #3: Yes

4. Is the manuscript presented in an intelligible fashion and written in standard English?

Reviewer #1: Yes

Reviewer #2: Yes

Reviewer #3: No

5. Review Comments to the Author

Reviewer #1: The manuscript is well written/However, some concerns should be addressed. Figures should be improved, especially the figure legends. Certain areas are in gray and the results should be discussed properly. See the attached review for further details.

Reviewer #2: 1. p. 7:33-34 - describe the solvent condition as ratio. for example, EtOAC : Hex 1:1-1:10 (v/v)

2. p. 7:38-39 - Spread the paragraph between method 2.6 and 2.7

3. I suggest that it is better to change the word : EtOH/Hex, EtOH/EA, EtOH/BuOH, EtOH/H2O to "Hex, EA, BuOH, H2O"

4. p. 10 Fig.2 - tag (a) and (b) for microscope data and migration graph each.

5. Fig.2, Fig.9(c) - In cell migration assays, change the labeling of y axis of the graphs: "relative cell migration (%)"

6. Fig.3, Fig.7(b) - change the labeling of y axis of the graphs : "relative intensity" or "relative intensity (fold)"

7. P. 16:4 - ***p<0.001, check the p-value.

8. Fig. 6 : your group suggest the Hyaluronidase inhibitiory activity of compounds as graph.

but same activity of extract and subfraction is presented as table (table 1). Unify the way you express data.

9. Fig. 7(c), your group suggest LMW-HA quantification data of extract, Hex subraction and compound 5 together.

but if there are no special excuses, I recommend your group to make figures for extract/subfraction and compounds each :

one data set of extract/Hex, EA, BuOH, H2O fraction and one set of compounds 1-10.

Afterwards, if placed in pairs with Table 1 and Figure 6,

it could additionally explain the role of hyaluronidases on LMW-HA degration.

10. Change the order of figure 8 and 9. Present in vitro assay at first and put insilico data later.

11. In Figure 8a, 3D image can show the binding of compounds more clearly.

Display Hyal-1 as 3D structure and change the direction that shows the binding of compound to the catalytic cleft more clearly.

I recommend this paper as an example : https://doi.org/10.3390/antiox13070767, figure 5A

12. write the PDB number. ;

Figure 8: a) Docking poses of quercetin (orange) and lutein in Hyal-1 (cyan, PDB ID: 2PE4)

Reviewer #3: In the manuscript "Lutein derived from Xenostegia tridentata exhibits anticancer activities against A549 lung cancer cells via hyaluronidase inhibition”, Chatwichien et al found the correlation why lutein can inhibit growth of A549 cells. Although the extract showed the antiproliferation and antimigration activities, the extract exhibited low anticancer potential, with the IC50 value > 250 ug/mL.

Sec 2.1. please indicate which part of the plant was extracted.

Sec 2.3. Please indicate hyaluronidase from which species.

Sec 2.6. Please indicate the type of silica gel column.

Sec 2.6. Figure 4 is first figure or remove “, as shown in Figure 4” (line 34, Page 7).

Sec 2.7. Supporting S3 is the first supplemental data mentioned in the manuscript.

Page 8, Line 34, where is figure 1c? The size of the figure should be the same.

Page 8, Sec 3.1., Anticancer should be fixed as cytotoxic. In addition, MTT assay is actually not the antiproliferative assay.

Sec 3.1. Which dataset was use to have selectivity indexes of 1.3 and 1.6? For concentration of log2.0, they were not different and the reviewer thinks whether this is correct?

It is very difficult to understand why the effect of the extract (EtOAc and BuOH) at concentration of 250 ug/mL is so poor, as compared the high effect at 100 and 50 ug/mL. In addition, the P value indicates significance.

Table 1. Quercetin concentration used is uM, not ug/mL, and thus, these results cannot be compared.

Fig. 8, hyaluronan and other compounds isolated in this study should also be docked into the enzyme as comparison. If kinetic data is present, it will be worse of determining the docking correction. Is it competitive inhibitor?

6. PLOS authors have the option to publish the peer review history of their article (what does this mean?). If published, this will include your full peer review and any attached files.

Reviewer #1: No

Reviewer #2: No

Reviewer #3: No

---

## [Author Response · Author response to Decision Letter 0]

19 Nov 2024

Reviewer #1: The manuscript is well written/However, some concerns should be addressed. Figures should be improved, especially the figure legends. Certain areas are in gray and the results should be discussed properly. See the attached review for further details. The study "Lutein from X. tridentata showing anti-cancer activity in lung cancer cells via hyaluronidase inhibition” is valuable and can be evaluated as potent to be published in PLOS ONE after major revisions. 

The clarity of the figures is improved. Also, the figure legends are revised to provide necessary information for understanding the figures. The results are further discussed. The details are as follows.

1. Though the introduction is comprehensive, only the given isolated compounds are the only flavonoids present in the studied plant. Also, it unclear whether the plant is used in Indian medicine as a single plant or as a polyherbal mixture. 

X. tridentata is a flavonoid-rich plant. According to the literature, the compounds isolated from the plant are mainly flavonoids. However, some other compounds have been also reported. Therefore, the sentence “Flavonoids, phenolics, ergosine alkaloids, pyrrolidine alkaloids like hygrine and nicotine have been isolated from the plant.” is added to page 6 line 4. 

The plant is used as a main ingredient in Prasaranadi Kashayam (Indian medicine). The word “main” is added in page 6 line 1 to make it clearer that the plant is used as a polyherbal mixture. 

2. It is better to use short sentences as it is too complicated to read at times. For example, page 5: lines 5-9 is one sentence.

The sentences are revised to be shorter and more concise.

3. The isolation method should be clarified further. It is also unclear why the authors used ethanol to for the extraction procedure. What was the final remaining extraction upon fractionation. The fractionation procedure should be clearly explained.

Ethanol was used as the extracting solvent due to its polarity and ability to efficiently dissolve a wide range of bioactive compounds. Also, it is an edible solvent and easy to remove. For the fractionation, the crude ethanol extract was added with distilled water before sequentially extracting with hexane, ethyl acetate and butanol, respectively. As a result, the final remaining fraction was the water fraction. 

The above information is added to the “Materials and Methods” and “Results and discussion” sections.

4. Abstract: The results should indicate anticancer activity. Give future direction using one line.

The anticancer activity is indicated. Future direction is added to the abstract.

5. Introduction: The authors should clarify is this the only mechanism by which NSCLC is upregulated. The authors should highlight the main components of the signaling pathway. It is unclear that the authors who studied, R. madagascariensis, managed to isolate a compound important for the inhibition. 

“In addition to other known cancer targets,” is added to page 5 line 1 to clarify that high level of HA is not the only mechanism involving with the proliferation of NSCLC. 

The study about R. madagascariensis was to investigate the hyaluronidase inhibitory activity of the plant extract. The authors found that the crude extract of the leaves exhibited potent hyaluronidase inhibition. Bioassay-guided isolation was also performed, yielding seven compounds identified as narcissin, rutin, epicatechin, isorhamnetin 7-O-glucoside, isorhamnetin 7-O-rutinoside, epiafzelechin, and kaempferol. The isolated compounds were tested for their hyaluronidase inhibitory activity. Results showed that epiafzelechin and epicatechin exhibited the highest inhibition values of 36.5% and 34.4%, respectively.

6. Materials and Methods

6.1. Give the coordination of the collected area. 

 The coordination of the collected area is given on page 7 line 11.

6.2. Give the temperature of the evaporator. 

 The evaporation was done at 40 °C under vacuum. The information is added to “Materials and Methods” section.

6.3. MTT assay – after how many hours of culture, the authors conducted the MTT assay. 

The cells were incubated with the extracts or the compounds for 48 h. The information is given in “Materials and Methods” section.

6.4. It authors could have conducted a siRNA inhibitory transfection to confirm the results of LWW hyaluronan. 

The siRNA experiment to silence the expression of hyaluronidase enzymes would be a good control to confirm the inhibitory activity of the extracts and lutein on hyaluronidase. However, since the technique is not our expertise, we performed the ELISA experiment to determine the level of LMW-HA in the culture medium of A549 cells treated with crude ethanol extract, hexane subfraction or compound 5, as compared to the vehicle control. (Section 2.6 and Fig 6c)

6.5. What was the column size used for isolation procedure.

The column size is added to “compound isolation” section.

6.6. Give details of using this particular protein for molecular docking (see Integration of in vitro and in-silico analysis of Caulerpa racemosa against antioxidant, antidiabetic, and anticancer activities. 2022 Sci Rep 12, 20848 (2022). https://doi.org/10.1038/s41598-022-24021-y

We have read the suggested article, followed this suggestion where applicable, and revised the section as followed: 

Protein and ligand preparation 

The crystal structure of human hyaluronidase 1 (Hyal-1) with a resolution of 2.0 Å was obtained from the Protein Data Bank (https://www.rcsb.org; PDB ID: 2PE4) [34] and downloaded using the accession number 2PE4. To prepare the protein for docking simulation, chain A of Hyal-1 was selected, bound ligands and water molecules were removed, using BIOVIA Discovery Studio Visualizer (2021) and saved in PDB format. The protein file was opened in AutoDockTools 1.5.6 and saved in PDBQT format. 

The selected ligands (compounds 1-10, quercetin, and hyaluronan) were drawn using ChemDraw Profesional 16.0, viewed on BIOVIA Discovery Studio Visualizer (2021), and saved in PDB format. The ligand files were opened in AutoDockTools 1.5.6 and saved in PDBQT format.

6.7. The statistical analysis should be more descriptive. 

The statistical analysis is revised.

7. Results and Discussion:

7.1. The authors should have used a standard drug to compare anticancer properties. Further, it is better to confirm antiproliferative assay using another technique.

Cisplatin was used as a positive control in the MTT assay. In this study, the MTT assay was employed as it is a well-established and widely used method to assess cell viability and proliferation in initial antiproliferative studies (References are provided below). The assay provides reliable data by measuring metabolic activity, which correlates with the number of viable cells. While additional techniques would provide further confirmation and mechanistic insights, this study is focused on the preliminary evaluation of antiproliferative effects. Further investigations, including cell cycle analysis and apoptosis assays, are currently underway in our laboratory to complement these findings and elucidate the underlying mechanisms.

Refs:

https://www.nature.com/articles/s41598-023-46867-6

https://journals.plos.org/plosone/article?id=10.1371/journal.pone.0078021

7.2. Data visualization could be improved.

The clarity of pictures and figure legends are improved. 

7.3. Fig 2. Photomicrographs are unclear. Introduce photos and give magnification.

The clarity of the figure is improved.

7.4. Obtained data should be discussed.

The obtained data is further elaborated upon, as marked by the yellow highlights in the revised manuscript.

7.5. According to the results, all the compounds have been isolated previously from other plants. Are there any new compounds isolated from the studied plant.

All the isolated compounds are known. There is no new compound obtained. However, this work, for the first time, reports the presence of fernenol, methyl-3,4-seco-8βH-fernadienoate, 2(4’-hydroxyphenyl)-ethyl behenate, 3,4-seco-8βH-fernadienoaic acid, lutein and glyceryl palmitate in X. tridentata.

7.6. Give the recorded therapeutic activities of the compound. When discussing, the interrelation the results are important. 

The recorded therapeutic activities of the isolated compounds are described in section 3.4. Flavonoids, flavonoid glycosides and phenolic compounds are known to exhibit moderate inhibitory activity on hyaluronidases, corresponding to our results. For most of the compounds isolated from hexane subfractions, very limited information on their biological activities has been revealed. For example, only antifungal activity has been reported for fernenol (1). The only report about 2(4’-hydroxyphenyl)-ethyl behenate (3) was its isolation from Buddleja cordata subsp. cordata and its moderated antituberculosis activity. The tetracyclic triterpenes, 3,4-seco-8βH-fernadienoic acid (4) and its methyl ester (2), were isolated from Euphoebia Chamaesyce. Compound 4 was found to possess stronger activity than its ester derivative, compound 2, on the antiproliferation against human cancer cell lines and DNA topoisomerase inhibition.

7.7. The docking analysis should discuss about van der Val bonding patterns and number of hydrogen bonds formed. See “In silico study of SARS-CoV-2 Spike protein RBD and human ACE-2 affinity dynamics across variants and Omicron sub-variants. Journal of Medical Virolog (2022). y. 94 (1), e28406 https://doi.org/10.1002/jmv.28406.

We have read the suggested article. The article refers to the van der Waals interaction as nonbonded interactions. Thus, we have followed this suggestion and added this particular discussion in the text as shown below and summarized the number of hydrogen-bond interactions in the revised Table 3:

Nevertheless, lutein (with 2 hydrogen bonds and 6 nonbonded interactions) exhibits significantly better in silico binding energy (Table 3) and better in vitro biological activities (Table 2) than both quercetin (with 0 hydrogen bonds and 7 nonbonded interactions) and compound 8 (with 7 hydrogen bonds and 5 nonbonded interactions). 

7.8. Also, Since the title is Docking Analysis, a much more detailed discussion is required in this section.

Although the word “Docking Analysis” was neither used in the title of this work, nor used in the title of the corresponding section and it was not originally meant to be the major part of this work, we have added a more extensive discussion based on the newly obtained data from additional molecular dockings. The additional data was summarized in the revised Table 3, presented in the revised Figure 7 and S4 Figure (supporting information), and a more elaborative discussion was added in the corresponding text as shown below: 

To understand the molecular interactions crucial for hyaluronidase inhibition, molecular docking studies of all of the isolated compounds (compounds 1-10) and hyaluronan were performed in comparison with quercetin [53–54] to analyze their binding interaction with Hyal-1, using the protocols as described in the ‘Materials and Methods’ section. The binding energies and amino acid interactions of the aforementioned molecules in Hyal-1 are summarized in Table 3. 

As shown in Table 3, the computational simulation results of compounds with highly favorable binding energies (compounds 5, 8, and quercetin) correlated well with the results from the inhibition assay (Table 2). Although compounds 1, 2, and 4 also showed favorable in silico binding energies, they exhibit relatively low in vitro inhibitory activity most likely due to their poor solubilities in the assay buffer, based on our observation. On the other hand, the docking of hyaluronan showed the expected relatively unfavorable binding energy, but the result may not accurately represent the actual binding pose of the hyaluronan with much longer chain lengths in Hyal-1 in nature. The rest of the studied compounds exhibit the in silico binding energies that roughly correlated with the trend observed in the in vitro inhibition assay. Since compound 5 (Lutein) showed the distinct efficacious in vitro inhibitory (Table 2) activity and in silico binding energy (-11.70 kcal/mol) compared to other compounds in this study, its binding interactions with Hyal-1 were further analyzed, in comparison with quercetin and other compounds. 

Among all of the studied compounds, compound 5 (Lutein) was the only compound that binds across the active site of Hyal-1 (from the Western region, across the center, to the Eastern region of the binding site, Fig 7a and b) [34], while the others either bind slightly above the active site (compounds 1 and 3; see S4 Fig), at the center only (compounds 4 and 9; see S4 Fig), at the Eastern region only (compounds 2, 10, and quercetin; see S4 Fig and Fig 7a and d), or around the center and the Eastern region (compounds 6, 7, 8, and hyaluronan; see S4 Fig). This optimal binding pose also made compound 5 the only compound in this study that interacts with both TYR75 and TYR210 (two of the residues on each end of the catalytic cleft responsible for the binding of sugar molecules) [34] via hydrogen-bond interactions by using the 3’ and 3 hydroxyl groups on both ends of the molecule (Fig 7c). Furthermore, this binding pose allows compound 5 to interact with TYR202, TYR286, and TRP321 (Figs 7a-c), which is located on the catalytic cleft of the active site of Hyal-1 [34], via hydrophobic interactions. On the other hand, in combination with other interactions, the interactions of quercetin with the important amino residues ASP129 and GLU131 (involved in the catalytic cleavage of the β(1,4)-linkages) [55] via π-anion interactions (Fig 7e) most likely explain the good in silico binding energy and the observed in vitro biological activities. Nevertheless, lutein (with 2 hydrogen bonds and 6 nonbonded interactions) exhibits significantly better in silico binding energy (Table 3) and better in vitro biological activities (Table 2) than both quercetin (with 0 hydrogen bonds and 7 nonbonded interactions) and compound 8 (with 7 hydrogen bonds and 5 nonbonded interactions). Based on our docking simulation, these observations are plausibly the result of the hydrogen-bond interaction of lutein with TYR75 (Eastern region) and TYR210 (Western region) combined with its multiple hydrophobic interactions with amino acid residues (TYR202, TYR210, TYR286, and TYP321) along the aromatic-residue-enriched sugar binding cleft [34] (Figure 7a-c), which is important for Hyal-1 activity [55]. These interactions ultimately contribute to the observed potency of this particular molecule. 

Table 3. Summary of in silico binding energy, inhibition constant (Ki), and amino acid residue interactions for compounds 1-10, quercetin, and hyaluronan. 

Compound Lowest binding energy [kcal/mol] with estimated inhibition constant, Ki Number of interacting amino acids (AA) and interacting amino acid residues with interacting distance (Å) Number of interactions (Number of Hydrogen-bond Interactions)

1 -10.9

(Ki = 10.3 nM) 3 AA = TYR84 (4.03), ASP129 (3.31, 3.19), ILE146 (4.41)

 4 (2)

2 -8.40

(Ki = 696.36 nM) 5 AA = GLU131 (3.33), TYR202 (3.77, 4.68), TYR286 (4.59), TRP321 (4.38, 5.52), TRP324 (4.35, 4.52, 4.78)

 9 (0)

3 -5.69

(Ki = 66920 nM) 6 AA = PRO62 (3.37), TYR75 (3.93), SER76 (3.28), TYR84 (4.53), ALA132 (4.01, 4.55, 4.89), ILE146 (3.75, 4.64, 4.82)

 10 (2)

4 -8.40

(Ki = 696.58 nM) 7 AA = ARG134 (4.12), GLY203 (3.26), PHE204 (3.85), ASP206 (2.95), TYR208 (2.67), TYR210 (4.94), PRO249 (3.84)

 7 (3)

5 -11.7

(Ki = 2.67 nM) 6 AA = TYR75 (2.08), TYR202 (4.49), TYR210 (1.92, 4.43), PRO249 (4.68), TYR286 (3.90), TRP321 (4.40, 4.87) 8 (2)

6 -5.29

(Ki = 132130 nM) 9 AA = ILE73 (3.44), GLU131 (3.60), ARG134 (3.57), TYR202 (2.68, 3.18, 4.74), GLY203 (3.31, 3.66), SER245 (3.00), TYR247 (4.89, 4.93), TYR286 (4.79), TRP321 (4.74, 4.74, 4.74)

 15 (1)

7 -7.99

(Ki = 1400 nM) 10 AA = ASN37 (2.74), GLU131 (3.77, 3.91), ARG134 (3.36), TYR202 (3.42), ASP206 (2.60, 2.86), TYR210 (2.85), SER245 (3.10), ARG265 (2.57), TYR286 (2.75, 4.96), TRP321 (2.79, 4.75)

 14 (8)

8 -9.10

(Ki = 135.3 nM) 10 AA = ASN37 (3.31), ASP129 (4.15), TRP130 (3.02), ARG134 (4.93),

---

## [Editor Report · Decision Letter 1]

28 Nov 2024

Lutein derived from Xenostegia tridentata exhibits anticancer activities against A549 lung cancer cells via hyaluronidase inhibition

PONE-D-24-42468R1

Dear Dr. Jaruwan,

We’re pleased to inform you that your manuscript has been judged scientifically suitable for publication and will be formally accepted for publication once it meets all outstanding technical requirements.

Kind regards,

Vinh Le Ba, PhD in Pharmaceutical Science

Academic Editor

PLOS ONE
---

## [Editor Report · Acceptance letter]

1 Dec 2024

PONE-D-24-42468R1 

PLOS ONE

Dear Dr. Chatwichien, 

I'm pleased to inform you that your manuscript has been deemed suitable for publication in PLOS ONE. Congratulations! Your manuscript is now being handed over to our production team.

Kind regards, 

on behalf of

Dr. Vinh Le Ba 

Academic Editor

PLOS ONE